# The Role of Context in Integrating Buprenorphine into a Drop-In Center in Kampala, Uganda, Using the Consolidated Framework for Implementation Research

**DOI:** 10.3390/ijerph191610382

**Published:** 2022-08-20

**Authors:** Julia Dickson-Gomez, Sarah Krechel, Dan Katende, Bryan Johnston, Wamala Twaibu, Laura Glasman, Moses Ogwal, Geofrey Musinguzi

**Affiliations:** 1Institute for Health and Equity, Medical College of Wisconsin, Milwaukee, WI 53226, USA; 2Uganda Harm Reduction Network, Kampala 31762, Uganda; 3School of Public Health, Makerere University, Kampala 7072, Uganda

**Keywords:** buprenorphine, syringe service programs, people who use opioids, opioid use disorder, Uganda, Africa

## Abstract

Background: Although Africa has long borne the brunt of the human immunodeficiency virus (HIV) epidemic, until recently, the continent has been considered largely free of illicit drug use and injection drug use in particular. In Uganda, the number of people who use or inject drugs (PWUD and PWID, respectively) has increased, and PWID are a key population at high risk for human immunodeficiency virus (HIV) and hepatitis C virus (HCV) infection. However, harm reduction practices, including providing clean injection equipment and medication-assisted treatment (MAT), have only recently been piloted in the country. This project aims to integrate buprenorphine into a harm reduction drop-in center (DIC). Methods: The Consolidated Framework for Implementation Research was used to guide our preparations to integrate buprenorphine into existing practices at a harm reduction DIC. We conducted key informant interviews with members of a community advisory board and DIC staff to document this process, its successes, and its failures. Results: Results indicate that criminalization of drug use and stigmatization of PWUD challenged efforts to provide buprenorphine treatment in less regulated community settings. Conclusions: DIC staff and their commitment to harm reduction and advocacy facilitated the process of obtaining necessary approvals.

## 1. Introduction

Although Africa has long borne the brunt of the human immunodeficiency virus (HIV) epidemic, until recently, the continent has been considered largely free of illicit drug use and injection drug use in particular [1]. Since 2008, countries in Sub-Saharan Africa have seen rapid increases in injection drug use, including injection of heroin and cocaine [1,2,3,4,5,6,7,8,9,10,11], resulting in a high prevalence of HIV and hepatitis C virus (HCV) infection, particularly in Tanzania and Kenya [1,12,13]. Drug markets have expanded west, and Uganda has experienced increasing numbers of people who use and inject drugs (PWUD and PWID, respectively) [14,15,16,17]. HIV prevalence among PWID in Uganda is estimated to be around 17% in multiple studies [14,17], with heroin being the most commonly injected drug among over 70% of PWID [18]. Further, most women who inject drugs also exchange sex for money or drugs, begin injecting earlier, and their HIV prevalence is likely even higher [18]. There are no current estimates of HCV infection as testing is not routinely done in Uganda, although HCV rates are high in other East African countries [13]. Qualitative research with 30 PWID in Kampala revealed substantial sharing of syringes and injection equipment and other risk behaviors such as mixing drugs with contaminated tap water and sometimes blood [19]. Drug overdoses were reported to be common. Sex exchanged for money or drugs was reported by all female participants, and HIV testing was reported to be infrequent [19]. In addition to the risk of infectious diseases, substance use disorders carry enormous health costs, including fatal overdoses [20,21], accidental deaths, violence [22], and other injection-related injuries such as sepsis, thrombosis, endocarditis, liver disease, and cirrhosis [23,24]. The social costs of SUD are equally devastating as people with SUDs are frequently incarcerated, un- or underemployed, or homeless [25,26].

Despite this serious risk, interventions to reduce the harms associated with injection drug use and drug-related sexual risk are just beginning to be developed and implemented in the region. In 2018, the Uganda Harm Reduction Network (UHRN) conducted the first pilot of a syringe exchange program in Kampala and is currently expanding this to other parts of the country. Although MAT has long been available in many countries in North America and Europe and is considered the gold standard for treatment of OUD [27,28,29], Medication Assisted Treatment (MAT) has just begun to be offered in East Africa. (While Medications to treat Opioid Use Disorder, MOUD, are now the preferred nomenclature for this class of treatment, MAT is still the most commonly used term in Uganda, so we will use MAT throughout the paper to avoid confusion.) However, even though both methadone and buprenorphine have been used for many decades in many countries such as the US, there is still a considerable amount of stigma regarding MOUD, which is amplified by tight regulations over its use [30,31,32,33,34,35,36]. Tanzania and Kenya recently opened methadone clinics in 2016 and 2018, respectively [37,38]. In Uganda, the first clinic to provide MAT was piloted in the national psychiatric referral hospital, Butabika, with funds from the President’s Emergency Program for AIDS Relief (PEPFAR) COP2018 and the Global Fund to Fight AIDS, Tuberculosis, and Malaria (Global Fund) key population investment. In addition to methadone, the Uganda MAT program distributed buprenorphine, a partial opioid agonist, which can be administered by physicians in outpatient settings rather than in tightly controlled clinics like methadone. The Uganda Harm Reduction Network was involved in the MAT task force in Uganda and in recruiting and referring PWID to Butabika for MAT treatment. However, patients on methadone and buprenorphine are required to visit Butabika daily for dosing, and no take-home doses are allowed. Further, Butabika, as a psychiatric hospital, does not offer other harm reduction services to PWID.

Integration of HIV prevention and care and MAT can have synergistic effects in preventing HIV infection and transmission [39,40,41]. HIV care has been integrated into methadone maintenance clinics in Tanzania [12,42,43,44], the US [45,46], and Ukraine [47]. Buprenorphine has also been integrated into HIV clinics in the US [48,49,50]. Patients receiving integrated care liked the convenience of co-located care [39]. Those who received antiretroviral therapy (ART) at MAT clinics reported differences in philosophy between MAT and HIV providers, with MAT providers generally expecting abstinence and having a punitive attitude toward those who do not achieve it [50]. HIV providers, on the other hand, were perceived by PWID as having a more client-centered and harm reduction approach [49]. Patients receiving ART from the MAT center in Tanzania reported fear of accidentally disclosing their HIV status to other patients if they received ART with their directly observed methadone doses [44].

While integrated services are advantageous compared to standalone services, there are barriers that may prevent PWID from accessing such services, including mistrust and stigma associated with HIV and injection drug use. Harm reduction agencies serve a diverse population of out-of-treatment opioid users with a variety of services, including the provision of sterile syringes and HIV/HCV testing. In spite of their large reach to out-of-treatment PWIDs and PWID’s stated preference for receiving buprenorphine in harm reduction agencies, only a few pilot studies have implemented buprenorphine in harm reduction agencies within the US [51,52,53,54,55,56,57,58], and none, to our knowledge, in low- and middle-income countries (LMICs). Offering buprenorphine in harm reduction settings not only may be more acceptable to PWID, but it can be offered to both those who are living with HIV and those who are not yet infected. Buprenorphine, in turn, has been shown to reduce injection risks, and to improve ART adherence [59].

Most researchers argue that a combination of approaches is needed to reduce HIV infection among PWID, including the provision of clean syringes through needle service programs (NSPs); frequent HIV testing and linkage to HIV care; antiretroviral therapy to reduce risk of onward transmission through viral suppression; MAT; psychosocial support and pre- and post-exposure prophylaxis [59,60,61]. In addition, modeling projections suggest that very high coverage of ART, NSPs, and MAT in combination are necessary to reduce HIV incidence of PWID by more than 50%; an even higher coverage of single interventions is necessary to achieve similar effects [59]. In line with these recommendations, in 2020, we received funding to pilot a program that provides buprenorphine treatment in one of the UHRN’s drop-in-community centers (DICs) in Kampala, Uganda, in order to increase accessibility to MAT and to allow the integration of MAT into a wide array of services to reduce the harms associated with injection drug use and prevent HIV in this community.

This paper analyzes the processes of adapting and implementing the buprenorphine delivery program at the UHRN DIC using the Consolidated Framework for Implementation Research (CFIR) [62,63]. CFIR is a framework that has been used in many implementation studies to adapt the implementation of an intervention to the local context [62,63] and was used as a guiding tool to collect formative research to integrate buprenorphine into the already existing services of a harm reduction drop-in-center. This paper draws on fieldnotes and minutes from the research team and community advisory board (CAB) meetings, and in-depth interviews with members of the community advisory board (CAB), UHRN staff, and other key stakeholders. CFIR describes five domains that may affect the implementation of interventions, and that should be considered when designing new or adapting existing interventions for implementation in community settings. These include (1) intervention characteristics (e.g., stakeholder perspectives of the value of the intervention, the complexity of the intervention, and essential or adaptable components); (2) the inner setting (e.g., factors within the organization that will be implementing the intervention that may affect its success, including human and physical resources, leadership engagement); (3) the outer setting (e.g., features external to the implementing organization that could affect success, such as policies, funding, political support); (4) the characteristics of individuals involved in implementation (e.g., their knowledge of the intervention and beliefs in its efficacy); and (5) the implementation process (strategies or tactics that influence implementation, such as identifying intervention champions or providing sufficient training).

## 2. Methods

A key liaison located in Kampala recruited and coordinated the program staff and the Community Advisory Board (CAB). Eight interviews were conducted over the Zoom meeting platform, ranging from 30 min to 70 min in length. These included all members of the CAB and all UHRN staff who were involved in the process of developing standard operating procedures and gaining approvals to provide buprenorphine. Interview length varied because participants had different levels of involvement in the process of adapting buprenorphine into the DIC. Participants included members of the CAB and UHRN staff. The CAB included a legal officer, a paralegal, and people with lived experience. Some UHRN staff attended the CAB meetings. UHRN staff interviewed included a medical provider, a social worker, and an administrative leader.

Interview questions covered the five domains of the CFIR (intervention characteristics, the inner setting, the outer setting, the characteristic of individual implementors, and the implementation process). Questions about the *intervention characteristics* included asking participants what they knew about buprenorphine before the project and their perceptions of its efficacy and safety. *Processes* of implementation were measured by asking how decisions were made and the processes by which the Standard Operating Procedures (SOPs) were developed. *Inner setting* questions evaluated the capacity of the DICs, including staff workflows and infrastructure capabilities. In addition, participants were asked to describe the mission of the DIC and how buprenorphine fits within it. To assess *the outer setting*, participants were asked to describe the standard operating procedure (SOP) development process, and stakeholder approval and involvement in that process. They were also asked about how people who use and inject drugs are seen by others in Uganda and how the social and legal culture of Uganda may specifically impact the provision of buprenorphine at the DICs. *Characteristics of the individual implementors* were assessed by asking participants about the kinds of training and experience they have and their feelings about integrating buprenorphine into their everyday work.

### Data Analysis

All interviews were recorded, then later transcribed verbatim. A preliminary coding tree was developed using two initial transcripts following the CFIR model. The preliminary list of codes was then applied to a subsequent transcript and further refined until consensus of the two coders was met. The finalized list of codes was derived from three transcripts that reflected the breadth of experience from participants (e.g., clinical, legal, and peer perspectives). Using MAXQDA software, the final list of codes was applied to the remaining transcripts. After coding, the interviews of people with different roles in the process (e.g., CAB member versus physician) were compared to determine whether there were differences in opinion based on their roles in the process. These codes were then sorted into the five CFIR categories.

## 3. Results

Results are organized according to the CFIR framework and presented in Table 1, along with the prevalence of participants who discussed each theme. Under “Intervention,” themes included stigma and misunderstanding of SUD and harm reduction, buprenorphine overdose and diversion concerns, and MAT stigma. Buprenorphine, and MAT in general, is a new intervention in Uganda, and so stakeholders and providers understandably had little experience with it and concerns about providing it in a community setting. Many of the concerns were the same as seen in the United States and other high-income countries, which have provided methadone for over 50 years, and buprenorphine for a couple of decades, and included fears of diversion and beliefs that MAT was “just another drug” that works in much the same way as heroin, and so people taking MAT have not achieved abstinence. These concerns must be understood in the larger Ugandan context in which illicit drug use is highly stigmatized and criminalized. On the other hand, all participants had favorable attitudes toward and believed in the efficacy of buprenorphine. The pilot program conducted at Butabika hospital addressed these concerns by being highly conservative in its approach, requiring patients to receive their doses of MAT daily at the hospital. According to clients of the UHRN DIC, this approach largely failed to treat any patients with buprenorphine and had low rates of retention in methadone treatment. The original Ugandan guidelines for providing MAT were seen as not taking the characteristics of PWID into account in the design of their SOPs. As a result, a newly developed UHRN SOPs allowed take-home doses of buprenorphine and home induction. The “processes” used to achieve these included much consultation with experts, including lawyers and members of various ministries, seeking buy-in and approvals, and using a broad-based and representative community advisory board to make decisions. UHRN and the research team sought and received approvals from a wide variety of regulatory agencies, including the Uganda National Council of Science and Technology, the Ministry of Health, the Centers for Disease Control and Infectious Disease Institute, and the National Drug Authority. The “outer context” influencing buprenorphine integration included the general context of criminalization, including the ambiguous legal status of buprenorphine, which was classified as an opioid and thus, illegal under most circumstances, and the legality of DIC provision of buprenorphine as it was not registered as a healthcare facility. CAB and UHRN staff were concerned that this legal ambiguity would lead police to arrest PWUID in treatment or shut down the DIC. Thus, they sought champions among police and policy makers to seek their buy-in. Counterbalancing the ambivalence of the Uganda regulatory agencies, Global Health Initiatives (GHI), such as PEPFAR and the Global Fund, have pressed for targeted prevention activities among key populations, including PWID. To receive funding from these, Uganda has had to start implementing harm reduction programs.

Many adjustments needed to occur in the inner context of UHRN as well, including obtaining more clinical staff, training them on buprenorphine treatment, and drawing on existing DIC strengths. Facilitating these processes was the belief that buprenorphine provision fitted well with the mission of UHRN and that the DIC was uniquely positioned as a known and trusted “safe space” to provide buprenorphine to PWUID. Ultimately, the UHRN wished to register their DIC in order to be able to provide holistic services, including buprenorphine, to their clients. UHRN staff saw engaging with regulatory agencies to get approval to provide buprenorphine in the community as giving them an opportunity to educate Ugandan officials about substance use disorders and people who use drugs and advocate for law and policy changes that they hoped would help to sustain and scale-up community provision of buprenorphine.

### 3.1. Intervention

Both illicit drug use and MAT are new in Uganda. While MAT is being provided for the first time, in Uganda, very few non-pharmaceutical drug treatments are available either, with Butabika being the only place that provided any kind of substance use treatment before the MAT pilot. A result of this lack of experience is that drug use is seen as a person’s moral failing, even among some medical providers.

People who use drugs in Uganda are considered a social immorality. We haven’t yet so much understood drug use as being a disease… In fact, we were punishing them for having chosen to use drugs… When someone gets addicted… and decided to quit and they quit, they need support. From all entities, they need support from the health workers… Most importantly, they need support from their families and their societies. Truthfully…, before I started working for Harm Reduction Network, I thought it was a matter of choice. (Interview 1) 

Harm reduction is also very new in Uganda and conflicts with the criminalization of drug use, as the participant below describes.

Culturally, we have here a culture where drug use is condemned. That the person who uses and injects drugs is looked at as a social misfit, is looked at as a social disgrace, is looked at as someone who should be locked away, someone who is not meant to be in society. Number two, we have a regressive policy, legal policy, regulatory framework. Very regressive. It is pro-criminalization… Number three is lack of appreciation of what the concept of harm reduction entails. So, to answer your question, you need to look into all those aspects, and they will inform you of the culture, of the environment, of the context under which these SOPs are going to operate. (Interview 4) 

As buprenorphine had only been provided in Butabika, piloting buprenorphine in a community setting was another large innovation that caused some unease along with excitement from UHRN staff and regulatory agencies.

One thing you have to appreciate is the fact… [that this] is the first time we have the medical assisted therapy in our country. So that means that it’s going to be the first time that we are having buprenorphine being administered from the community level. But also, it’s going to be the only center of buprenorphine administration from the community. There’ll be two centers, that is Butabika which is not so much giving out buprenorphine for reasons of the protocol that they have to follow, and then Uganda Harm Reduction Network, so the context is when it’s new, it’s all new to us, so we shall be doing a little benchmarking, we should be doing a lot of consultation… because we are… are very shy, yes… but the fact that we have people that we can relay… or… consult [if] we are worried about anything, there is no panic so it’s very good. (Interview 1) 

Butabika is on the outskirts of Kampala, far from where most PWID live. In addition, Butabika’s protocol alluded to above, requires people to come in for daily dosing for methadone and buprenorphine. This, and the requirement to make patients wait in the hospital to be inducted, has led to few patients being started on and maintained on buprenorphine (3 to date). Further, the distance needed to travel for daily dosing has led many PWID to drop out of or be expelled from the program.

Methadone is centralized and it’s done in Butabika. And the fact that Butabika is not in a central area. It’s a bit far away from the town, the capital city. So there have been challenges of transport to… be able to reach Butabika. And the fact it’s a daily dose that has to be gotten from the clinic, that every single day a person has to move to Butabika. It has been a big challenge and has caused a little follow-up issues. And so many people have dropped out of the program, not so many, but there is a percentage of patients that have dropped out of the treatment because of the fact that it has to be on the daily to go to Butabika to get medication. (Interview 6) 

However, take-home medication increased concerns about diversion, misuse, and potential overdose.

Then my other concern was in regard to overdosing and getting addicted to buprenorphine and my concerns were laid to rest… But it also in regards to overdose yes, because it has a [naloxone] it’s very hard for someone to overdose. (Interview 1)

[Buprenorphine] can even go in a community. They give it to like you know mostly girls who would never do drugs. Someone tries to give them buprenorphine you know. Or you can leave it, like some of them they have children home. You can leave it there, then his children try to swallow it. You know what can happen. (Interview 7) 

The efficacy of MAT was seen in the Butabika pilot, and it was hoped that by increasing access, more PWID could benefit from it. Participants also saw the ability for buprenorphine to be taken at home as a great advantage over methadone. However, among key stakeholders, there was still ambivalence toward implementing this approach, which was also associated with the belief that buprenorphine is “just another drug” and that taking it prolonged addiction.

There’s a good thing because I have seen some of my community members, some of them they have changed like the way they were being. It’s not like the same way they are living right now. Some of them they are started and connecting with their family. Some of them they have started doing some work, now they can get jobs where they used to work. (Interview 7) 

The challenges that came up, … the fact that the government some actors in the government think that we are giving persons who use and inject drugs more drugs in the form of what they’re using. Because they don’t understand the costs of using drug use and the community psycho-social economic support for persons on drugs, that was the cause them to replace. (Interview 4) 

Part of the reason that MAT was seen as just another drug was that it needs to be taken for an extended period and does not actually provide a “cure” for opioid use disorder, a concern that was mentioned frequently in early research team meetings.

### 3.2. Processes

Given that buprenorphine was new in Uganda and that the SOPs UHRN DIC was proposing were more flexible than Butabika, the UHRN brought together diverse perspectives for the CAB including legal experts, DIC staff, and people with lived experience.

Okay yeah so, this being a study, the initial stages included consultation with a couple of stakeholders to identify what we probably need to get to have such a drug at the drop-in center. And then, following the consultation with different stakeholders getting their buy-in and then understanding and seeing how this is through consultations with that team from [MCW]… in regards to administering buprenorphine from the community, and we build to that to suit our country’s context… However, with the development of that CAB as well, and the constitution of the CAB including almost all fields, the law enforcers, human rights advocates, the peers from the peer educators, that the educators from the community of people use drugs… We managed to also consult the policymakers in this which includes the National Drug Authority or now, we can have buprenorphine in the community, because that drop-in center is considered as community. (Interview 1) 

As the participant above mentions, approvals needed to be sought from various regulatory agencies in addition to the CAB.

In our community advisory board structure, we have even the community people…, people who use drugs community… So, we got a challenge whereby we had to have their input but the issue of being technical, the issue of language… We so much had to explain objectives… why are we developing this so that we could get each and everyone’s input. Then the other issue was that in order to have the final documents, we need even to consult the Minister of Health. You know how bureaucracy [is] in the government structures. You have to go this one and this one. It was in draft for some time. But at the end of the day, we had to find ways of finalizing them. So, the other thing is putting competing priorities. People had a number of priorities which all when we are developing this had to be included. (Interview 3) 

The various regulatory hurdles that needed to be addressed are discussed next as part of the outer context.

### 3.3. Outer Context

As mentioned above, the criminalization of drug use conflicts conceptually somewhat from harm reduction practices and many see harm reduction as promoting drug use.

For every crime that is committed in Uganda currently it’s… you as a drug user… So those elements now for drug users in Uganda, you are naturally a criminal, whether we committed it or not. But by the fact that we’re using drugs you’re considered to be a criminal. So, to us this side, of course UHRN excluded, … there’s no need for you to have any supportive laws. There’s no need for you to have any intervention that is going to help you. The only answer for you is incarceration because all crimes are attached to drug users. So, what does that mean? If there’s no supportive laws then it means even anyone who would like to come up with an intervention geared towards helping a drug user, it is looked at as promotion… Just like the NSP program it is still done back doors. Like for someone to embrace it, that NSP is actually looking at reducing the rate of transmission or any other blood borne infection due to the sharing habits that PWIDs usually have… I don’t know why they are not looking at NSP like condoms because, if condom is to prevent pregnancy and any other STI, then why not also NSP… So, the element of criminality is the one that is affecting most of the programs that would have come to place to help drug users actually live a healthy life. (Interview 2) 

In addition, buprenorphine is currently a classified drug, and only medical personnel are allowed to prescribe or dispense it, and only people with prescriptions are allowed to possess it.

Now, the people exempted from this, listed under sub section 3, a person who has possession of narcotic drug or psychotropic substance under a license issued under section 27 of the National Drug and Policy Act permitting him or her to have possession of the narcotic drug or psychotropic substance. Now section 27… talks about possession of classified drugs. So, the exception under which you can have that drug is when… you have a permit under the National Drug Authority: Part B, a medical practitioner, dentist, veterinary surgeon, or registered pharmacist, who is in possession of a narcotic drug or psychotropic substances for any medical purpose; C, a person who possesses the narcotic drug, psychotropic substances for a medical purpose from or using prescription of a medical practitioner, dentist, or veterinary surgeon; D, a person authorized to be in possession of the drug. So, the only circumstances under which the client can be in possession, is when they have prescription from a medical practitioner. To answer your question, however, I put a disclaimer. Due to the gross human inhumane way persons who use and inject drugs are treated during operations and during arrests, I have serious doubts that this exception or having in possession of narcotic drugs will be considered effectively to prevent arrest of that [inaudible]. (Interview 4) 

Because of this, members of UHRN were concerned that participants who were provided take-home doses of buprenorphine would be arrested. To forestall this, the CAB included law enforcement, lawyers, and paralegals and gave a special presentation to police officers in the district.

The main issues were always around how we are going to be dispensing such a highly controlled drug from that drop-in center and that was addressed then to how we are going to let the people who are using and injecting drugs have such a drug in the community? How are we going to avoid they’re sharing of this medication? What are we putting in place to ensure that actually that people that are supposed to be taking a medication are the ones that are taking this course, remember with that the MAT that is being offered at Butabika are directly observed. But then, in this case, we are giving people medication to take home, giving them doses. So, we had issues with law enforcement in regards to having updates in the community, I know we’re going to control that but we assure them [police officers] that will be working closely with peer educators… The major, major push that help us go beyond all those barriers was the fact that we have with secure champions within the law enforcement agencies people that can pass on the right information, but then also engaging them in other systems meetings that we’ve had in Uganda, so they have a wider understanding of this. (Interview 1) 

The National Drug and Policy Act quoted above complicated allowing the DIC to prescribe and dispense buprenorphine since, in Uganda, in order to prescribe medications, an organization needs to be registered as a medical clinic. To be registered, the clinic has to fulfill certain requirements such as the number and kinds of medical staff:

Our DICs, those are the drop-in centers, have not been accredited by the government of Uganda, though we are in the process. So, the DICs, because they are not accredited, are not mandated to give treatment like counseling, psycho-support…and that sort. So, because of that, when they hear about dispensing buprenorphine from the DIC, people you know think the other way, but I think we are able to bridge this. Why? Because of the team I have talked about. We have the Minister of Health on board, we have the research team, academia on board. So, I know we will try to bridge that gap. (Interview 3) 

The process of registering the DIC, however, met with some delays, and so UHRN sought approvals from a number of different regulatory agencies and medical providers to obtain permission to prescribe and dispense buprenorphine and to allow home induction and take-home doses.

There were quite a quite many [approvals needed]… The highest authorities in the Ministry of Health. Then also the medical personnel, the approvals from the district medical personnel.

Interviewer: Were there difficulties in obtaining these approvals?

Interviewee: Yeah… Approvals like this… going into different offices. Also, many times you find the right offices and finding they are not available. And, also, the financial strain whereby you had to use some… money to make sure you move to and from… And also, the other ability of the officers. The right officer… It was difficult getting them. (Interview 5) 

As the CAB member below described, difficulties in obtaining the necessary approvals were partly a result of the criminalization and stigmatization of drug use which made many officials suspicious of MAT and harm reduction as an approach.

Basically, … the approvals wouldn’t be bad, and they wouldn’t be a negative thing if there are no problem attitudes and stigma associated with drug use. That’s what makes everything difficult because very few people who are meant to approve such procedures appreciate the concept of harm reduction. Most of them are pro-criminalization of drug use… It creates discomfort to approve because its attitude is criminalization, its attitude is not pro-harm reduction. (Interview 4) 

Counter-balancing these stigmatizing attitudes toward PWID, was the fact that they were members of the Global Fund’s key populations. The UHRN was able to advocate for community buprenorphine treatment and needle exchange by explaining that providing buprenorphine to more PWID would help reduce the transmission of HIV and improve HIV morbidity and mortality for PWID.

The Uganda Harm Reduction Network has, we have positioned ourselves in a way that we do a lot of collaboration and networking. So, the fact that we already had understanding, most of these agencies or most of our stakeholders, it was pretty much easy to have … the buy in… I can’t say it’s been a walk in heaven but the challenges were very easy to, you know, go past because these are people that have already understood that kind of work we are offering one, but it was voted to appreciate drug use as a public health problem… a social problem… because HIV goals that by 2030… we cannot leave people who use and inject drugs behind so when we’re selling to them or telling them MAT is actually something that can help in this fight against HIV, the fight against the blood borne diseases amongst people who use and inject drugs. It can help into getting people using drugs into a better place reengage them. (Interview 1) 

### 3.4. Inner Context and Persons to Deliver the Intervention

To be approved to provide buprenorphine, the UHRN DIC had to make some changes in its staffing. Some of these changes were necessary to become registered as a clinic, while others were made to be able to provide buprenorphine while waiting for the DIC registration to be approved. This required agreeing to supervision by clinicians at Butabika and the Medical College of Wisconsin.

Yes, so now looking into that… establishment of the drop-in centers in Uganda basically we shouldn’t be able to give out any medication to the STI treatment and even when we are initiating at PrEP we need to be having support from the health facility. Now, when we mentioned to them that there is a full-time clinician… registered with that Lighthouse Counsel, which is the regulating body for health workers allied health professionals in Uganda, that provided a safe space for them to actually trust that dispensing of this medication from the community. But also assuring them that we shall be working closely with all the referral hospitals and also that public hospitals that we are already partnering with them, already have a memorandum of understanding with them, just in case of any situation or any adverse effects, any unexpected events we can always have support… Then they could let go of that sector in the drop-in center SOPs and allow us to carry it. (Interview 1) 

The UHRN had to increase the number of clinical staff in the DIC to fill the requirements to register it as a clinic.

We [were] also advised we had to increase the number of staff because the dispensing this medicine involves a lot of handling in terms of the staff that can guide the clients after taking their doses. That meant we had to add in a few staff in terms of we added another clinician that is going to act as a counselor. And also, during that training we had to train the administrator officer who is also going to support that team. (Interview 3) 

The UHRN clinical staff also received training in inducting and maintaining patients on buprenorphine from a family practice physician with considerable experience treating patients with buprenorphine. Although training was held online due to restrictions related to the COVID-19 pandemic, training was held over several months, and clinicians were able to focus on areas they felt uncertain about by requesting these in advance.

[The doctor] was kind enough to ask us where we thought he would put more emphasis, or which topics were missing… so that training was really comprehensive [at the] outset… It was very detailed and it was flowing like it was very hard to miss a point. So, it was very, very comprehensive, but I also appreciate it, because we’ve had a couple of trainings in other fields, but I don’t think we’ve had our training, where people get to ask you where you feel we should put more emphasis. (Interview 1) 

Staff were also carefully consulted when developing the SOPs to make sure that there were sufficient human resources to cover the work, and to fit buprenorphine treatment seamlessly into other UHRN activities.

First of all, they considered the availability of the human resource that is going to actually run this whole buprenorphine program. At what stage it was, particularly, of course, the key elements that were looked at as a clinician, who is available? Then the counselor, who is also available? And then also these other elements of that case manager and then the also at the Community level, the mobilization and all that. And the linkage facilitator will do the escorted linkage and referrals and all that. So, as we are looking at the SOPs, they are looking at their availability of all of that, human resources that is available on the ground. And the capacity that they are going to have, or the workload is currently because that is your question. The continuous question our boss keeps asking us. (Interview 1) 

UHRN leadership and frontline staff were highly committed to providing buprenorphine at the DIC because they saw MAT as an essential tool in harm reduction practices. The needs and experiences of PWUD is the center of harm reduction, which was described as the UHRN’S mission.

Our vision at UHRN, we want to see a community of people who [are] using drugs free from HIV, from HCV, free from HIV and HCV. Now, when we are offering buprenorphine, buprenorphine makes people sober, makes people who use drugs, in other words, let me use a stigmatizing tongue, come back to normal. And when they come back to normal…they can be able to have agency over their lives and at the end of the day, for those for example who have been injecting and they didn’t know the issues around injecting, they come back to learn the… [risks of] sharing [injection equipment]… The HIV patients who are using drugs can be able to take their medicines. Then even the women who use drugs can even be able to access the STI [sexually transmitted infection] treatment. So, there is that link… Then you also talk about our mission. And in our mission, we are looking at providing a comprehensive service to a person who [is] using drugs and when you look at buprenorphine, the MAT component was missing in Uganda. So, if we have it, then we feel that greatly contributes to the HIV response and hepatitis response in Uganda among people who use drugs. (Interview 3) 

The UHRN provides not only HIV prevention materials, including PrEP and ART but they also help with other psychosocial problems the PWUD may have. This holistic vision of interacting with PWUD requires non-clinical staff.

I think the [DIC has a] very good staff structure for offering harm reduction interventions considering all their needs… We have two clinicians, then we have social workers, we have offices that can offer access to justice. It’s a whole package. It’s almost a self-sustaining but then we have the research and documentation department is really very detailed and broad. It has very professional researchers and then research assistants, so we didn’t have to make very many adjustments when it came to the staff structure. (Interview 1) 

Providing harm reduction services also helped to engender PWUD’s trust in the UHRN DIC and its staff, something that participants noted was absent at Butabika.

The fact that this DIC has been in existence since 2008 and these people are already coming around and receiving different HIV prevention services, before even methadone came into the country. So, they are already used to the place. And the fact that it’s also centrally located in the community, they find it very accessible. And the fact that they were already coming to receive HIV services. So, them coming again for buprenorphine, for MAT, is not something that is different or new to them because they are already used to the place, and they are used to the team that is already here. They are already feeling at home. (Interview 6) 

The needs of PWUD were taken very much into account in developing the SOPs at the UHRN DIC. The UHRN staff conducted focus groups with PWUD who were referred to MAT treatment at Butabika and were not inducted or dropped out and used these lessons to increase the accessibility of buprenorphine.

Of course, the DIC being in the community arm, every intervention that comes… it has to be community centered. So, if we are looking at a program that has come in to help to solve a problem that the community is facing, and then it is the very program that is making the community’s life hard or complicated then we find that we have different interests, as we are serving the same population.

So, as the DIC we ensure that at least the program is going to benefit the community. Of course, we shall have standard operating procedures, all the guidelines to follow as we’re implementing but, at the end of the day, you are not treating the SOP. It’s just a guideline to where and what to follow and then at the end of the day, you are looking at a client who has come to get changed. And then that means whatever intervention or whatever response you’re going to do to render it has to be to benefit the client, because you can’t say I disqualify you from treatment, treatment that we know is supposed to be for daily dose, we disqualify him because you threaten to beat us. Or disqualify you from treatment because you’re overly using alcohol. I mean this person has come here because they have a problem, and they need to change. So, it is from this point from the DIC that they have to get that change and not sending them back to the community to get changed and then come back. (Interview 2) 

To provide the more patient-centered buprenorphine treatment, SOPs that differed from those developed for the pilot MAT conducted at Butabika needed to be developed and approved by the various regulatory authorities described above. Rather than seeing obtaining approvals from various organizations as a burden, however, some participants saw it as an opportunity to advocate for changes in laws and policies that marginalize and criminalize PWUD and increased funding and support for harm reduction.

Maybe the other one we still need the advocacy to make the government embrace such initiatives. The recent study we conducted of assessing, it was called the finance assessment. Assessment of investment. And investment in harm reduction, we found out, we found out that this project, this research was funded by Harm Reduction International… not even an amount but any kind of in-kind support that we get form the government of Uganda for harm reduction. The rest is from external people…. That means we still need more advocacy for the government to embrace harm reduction and such interventions like medical assisted therapy or buprenorphine. And… lastly, we still need the research. Clinical trials in harm reduction. The only research we have that brings out the numbers of people who inject drugs is that national size estimates that was conducted by UNAIDS and commission. And still it relied on grey literature and have not gone to do real research to find out the size estimates of people who use or people who injected drugs in Uganda. (Interview 3) 

## 4. Discussion

This paper used implementation science, and CFIR in particular, to understand the process of implementing a novel intervention, MAT, to reduce opioid use and associated HIV risks in the Ugandan context. Results highlighted the importance of understanding the social and political context in implementation efforts. In bringing buprenorphine treatment to community settings, recently developed regulatory frameworks had to be adjusted. The original MAT guidelines and adjusted community guidelines occurred in the context of criminalization and stigmatization of drug use. As a result, the guidelines for the initial pilot of buprenorphine in Butabika were very conservative, requiring on-site buprenorphine induction and daily observed dosing. In this context, the UHRN’s efforts to bring buprenorphine to community settings were met with caution and approvals from various regulatory agencies were needed. In addition, approvals were obtained only for the purpose of research and the UHRN will need to obtain more approvals to continue the work if the pilot is successful. The time-consuming process of obtaining approvals and adjusting protocols was tolerated by the UHRN staff because they saw providing MAT as central to their mission to meet the needs of PWID and because they saw these efforts as helping to educate decision makers and advocate for PWID’s rights. The UHRN brought key stakeholders into the process of building SOPs from the beginning, including decision makers and people with lived experience. It has become more apparent in recent years that politics and public health are intimately linked, as the past COVID epidemic has proven. Learning from the strategies, successes, and failures of implementing novel interventions, therefore, can help to illuminate more and less successful implementation strategies.

Experiences from the US in providing buprenorphine in harm reduction drop-in-centers or mobile health centers show some similarities and differences due to the different contexts [53,54,55,56,57]. PWUD who use NSPs were enthusiastic about having buprenorphine provided in the drop-in centers and pilots of these have shown that providing buprenorphine in DICs is both acceptable and feasible. However, a qualitative process evaluation of eight NSPs that were funded by the New York City of Health and Mental Hygiene revealed various programmatic barriers, including staff knowledge and comfort with buprenorphine, hiring buprenorphine providers, and financial constraints [55]. In particular, finding providers who were knowledgeable about harm reduction and comfortable working in the setting, and who could also provide malpractice insurance was difficult. NSPs could not afford to obtain malpractice insurance themselves [55]. Several projects quickly reached treatment capacity due to limits in the US on the number of patients on buprenorphine a provider can treat [53,54,55,56]. Increasing staffing and training providers and UHRN staff on buprenorphine was also needed in our project, and various regulatory barriers needed to be overcome, although patient limits and malpractice insurance was not one of them. The regulatory barriers involved in integrating buprenorphine into harm reduction centers may be one of the reasons for the limited number of programs that have provided these. However, at the same time, the limited research in the US and our research in Uganda has shown that integrating buprenorphine into harm reduction services often requires engaging with policy makers that can ultimately serve to change policy to make provision of MAT in nontraditional settings easier.

The tension between harm reduction and criminalization of drug use is not unique to Uganda. Although the World Health Organization, the United Nations Office on Drugs and Crime and UNAIDS described and secured high-level political support for a comprehensive package of harm reduction interventions, including providing clean injection equipment and MAT, coverage remains low and such efforts are under-funded [64]. Between 2002 to 2014, the Global Fund invested $620 million for harm reduction in 58 countries, far short of the estimated $2.5 billion needed in 2015 alone [65]. It is widely recognized that an enabling legal environment in necessary to allow harm reduction policies room to function; however, harm reduction is still considered politically sensitive and morally unacceptable in many countries [64]. In many places, a legal environment enabling harm reduction exists side by side with continued criminalization of people who use drugs. For example, while Mexico passed the *Narcomenudeo* drug law reform to allow a more health-based approach to substance use in 2009, local police roundups of homeless PWUD continued. Police arrested PWUD for “quality of life” crimes such as loitering or tried to convince them (sometimes coercively) to enter treatment. These arrests displaced many PWUD, disrupted their networks, and increased their overdose and injection risks [66]. Similarly, while drug use was de-criminalized in Vietnam, police still used their discretion to take PWUD to compulsory drug treatment centers [67]. Further, research has consistently shown that even when allowed to operate, an environment which criminalizes drug use may discourage use PWUD to use harm reduction services, fearing or experiencing surveillance and arrest [66,68,68].

## 5. Conclusions

Global HIV/AIDS donor organizations have helped overcome some of the inertia of many governments in low- and middle-income countries toward harm reduction, and this is also seen in Uganda. The Global Fund has increasingly focused on providing services to “key populations” with the highest rates of HIV infection, including PWID [65]. An early NSP pilot at UHRN and the MAT pilot at Butabika were both funded by PEPFAR. Global Health Initiatives can exert pressure on governments to provide harm reduction services. Given the difference in policy on paper and as implemented in practice, however, it is necessary to highlight implementation problems and gaps. Thus, in addition to providing evidence that harm reduction efforts are effective at reducing the harms of drug use, research is also needed on the implementation of harm reduction strategies to hold governments and global health initiatives accountable and responsive to the needs of PWUD.

## Figures and Tables

**Table 1 ijerph-19-10382-t001:** Prevalence of themes in CFIR domains.

Themes	Prevalence
**Intervention**	
Stigma and misunderstanding of SUD and harm reduction	100%
Concerns about overdose and diversion	75%
MAT stigma	38%
Perceptions of MAT efficacy	100%
**Processes**	
Consultation with experts	75%
Seeking buy-in/approvals from stakeholders	88%
Community advisory board	63%
**Outer setting**	
Criminalization	
Legal status of buprenorphine	63%
Legality of DIC to provide buprenorphine	75%
Advocacy with police and policy makers	100%
Pressure from GHIs	13%
**Inner setting**	
Human resources	
Staffing needs	75%
Building on existing strengths	75%
Training	63%
Fit with UHRN mission	75%
DIC known and trusted by community	75%

## Data Availability

Interview transcripts can be obtained by request from the first author, jdickson@mcw.edu (J.D.-G.).

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
