# Peer review of "The Role of Context in Integrating Buprenorphine into a Drop-In Center in Kampala, Uganda, Using the Consolidated Framework for Implementation Research"

_ijerph, 2022, doi:10.3390/ijerph191610382_

Round 1

Reviewer 1 Report

Given that MAT initiatives are used elsewhere in the world I am surprised that this is not mentioned in the Introduction section. The success or not of these and other related ventures in Switzerland, the UK and other countries should be noted and would support the case for it's introduction for similar reasons in Kampala Uganda. Whilst the focus is buprenorphine as an opioid useful references which can direct reading can be found here  Diamorphine shortage could be more than just a problem for drug users (theconversation.com) 

The context of the new scheme is very well described but to the detriment of not expanding our knowledge re the global phenomenon of drug misuse and the negative impact of this on people and society. Also when you mention phrases such as sex exchange for the first time please make it clear what you mean. I guessed but I could have got it wrong.

Further, most women who inject 35 drugs also participate in sex exchanges, begin injecting earlier.....

The justification for the use of the CFIR is not clear. It is a very useful model in analysing and guiding intervention implementation but the authors should present this case. 

8 interviews were conducted - would you say that this number recruited would give a good insight into the phenomenon? So the views from 8 would generally represent the larger group? 

Can you make it clear that the inner setting and processes focus on SOPs is different as it does read that they are similar.

The data analysis phase as described is appropriate. If the final set of codes were derived from 3 transcripts can you advise on the value of the remaining 8 transcripts? 

Results - Pg 4 you mention the USA in your results and this was something I expected to see reference to in your Introduction section. Examples of such schemes in other countries. I still feel this is necessary to fully analyse your findings. 

This is an interesting finding - As a result, a newly developed UHRN SOPs allowed take-home doses of 167 buprenorphine and home induction - you may want to review the old traditional method of diamorphine use in the UK which was self injection at home (see the article as linked earlier). 

It would be helpful if it was clear where quotes start as they are not denoted by italics or quotation marks. I appreciate that this is the journal format and not the authors. 

Please ensure that anonymity is preserved in this section and names are changed to ensure participants are not recognised. 

Your results are quote heavy, with large quotes in the main being the results. It would be better to have a balance of text supported by quotes to amplify a point and bring it to life. Currently it reads as the other way round. You have good information here, so it would be good to discuss your findings and then analyse accordingly. 

Each of the CFIR elements has between 2-4 respondents, is this enough to get a flavour of the implementation issues, barriers, influences etc? You had up to 8 respondents to refer to so why is there not greater representation in the results? 

Discussion - I would have hoped to have seen more critical discussion in this section. If the literature was extended in the Introduction this would have been picked up again here to really understand what this study offers by way of intervention for this patient group versus similar initiatives in the world. There is a lot of reported quotes in the results section which can be more targeted and free up words/space for more discussion. 

This is a really interesting study which unfortunately reads as quite descriptive. I do think more can be done to make it a more critical appraisal of what this study shows us about the MAT schemes and its analysis using the CFIR and would urge the authors to reflect on this. 

I would encourage you to introduce new literature in the Introduction section that can be reintroduced to explain your findings. Be more critical in your appraisal of what you did and how it was implemented in the Discussion section and reduce the quote content as it has taken over your results section. 

Author Response

Thank you for your careful review of our manuscript.  I outline the changes to the manuscript below.

Given that MAT initiatives are used elsewhere in the world I am surprised that this is not mentioned in the Introduction section. The success or not of these and other related ventures in Switzerland, the UK and other countries should be noted and would support the case for it's introduction for similar reasons in Kampala Uganda. Whilst the focus is buprenorphine as an opioid useful references which can direct reading can be found here  Diamorphine shortage could be more than just a problem for drug users (theconversation.com) 

We have added a statement that MAT has long been provided in North America and Europe and has long been considered the standard of care, that it has only recently been provided in Africa.  We also argue that while MAT has been available for many years in many countries, it’s use is still highly stigmatized.  We also added references to indicate that providing buprenorphine in harm reduction settings (syringe services programs) is still relatively rare, even in settings that have long provided MAT.  We feel that this is what makes our project unique.

“Although MAT has long been available in many countries in North America and Europe and is considered the gold standard for treatment of OUD [27-29], Medication Assisted Treatment (MAT) has just begun to be offered in East Africa.  (While Medications to treat Opioid Use Disorder, MOUD, are now the preferred nomenclature for this class of treatment, MAT is still the most commonly used term in Uganda and so we will use MAT throughout the paper to avoid confusion.)  However, even though both methadone and heroin have been used for many decades in many countries such as the US, there is still a considerable amount of stigma regarding MOUD which is amplified by tight regulations over its use [30-36].”

The context of the new scheme is very well described but to the detriment of not expanding our knowledge re the global phenomenon of drug misuse and the negative impact of this on people and society. Also when you mention phrases such as sex exchange for the first time please make it clear what you mean. I guessed but I could have got it wrong.

Further, most women who inject 35 drugs also participate in sex exchanges, begin injecting earlier.....

We changed the phrasing of this to say most women “exchanged sex for money or drugs.” We also added literature about the negative effects of substance use to people and society in addition to hepatitis and HIV in the first paragraph of the introduction.

The justification for the use of the CFIR is not clear. It is a very useful model in analysing and guiding intervention implementation but the authors should present this case. 

We added a sentence justifying our use of the CFIR to integrate buprenorphine into the services provided by the UHRN drop-in-center. 

“CFIR is a framework that has been used in much implementation research to adapt implementation of an intervention to the local context [55, 56] and was used as a guiding tool to collect formative research to integrate buprenorphine into the already existing services of a harm reduction drop-in-center.”

8 interviews were conducted - would you say that this number recruited would give a good insight into the phenomenon? So the views from 8 would generally represent the larger group? 

The eight individuals interviewed included the entirety of the group in Uganda who were involved in the process of integrating buprenorphine into the UHRN.  This is now clarified in a statement in the results.

Can you make it clear that the inner setting and processes focus on SOPs is different as it does read that they are similar.

They are similar but “processes” reflects how decisions are made within the organization while inner setting focuses on existing resources and capacities.

The data analysis phase as described is appropriate. If the final set of codes were derived from 3 transcripts can you advise on the value of the remaining 8 transcripts? 

All eight interviews were coded so that they could be compared to determine if there were differences in opinion on any of the domains based on participants’ roles in the process.  Thus, we added the following statement.  “After coding, the interviews of people with different roles in the process (e.g., CAB member versus physician) were compared to determine whether there were differences in opinion based on their roles in the process.” 

Results - Pg 4 you mention the USA in your results and this was something I expected to see reference to in your Introduction section. Examples of such schemes in other countries. I still feel this is necessary to fully analyse your findings. 

We added information about MAT programs in the introduction (see above) and also added a statement regarding stigmatization of MAT. 

Although MAT has long been available in many countries in North America and Europe and is considered the gold standard for treatment of OUD [27-29], Medication Assisted Treatment (MAT) has just begun to be offered in East Africa.  (While Medications to treat Opioid Use Disorder, MOUD, are now the preferred nomenclature for this class of treatment, MAT is still the most commonly used term in Uganda and so we will use MAT throughout the paper to avoid confusion.)  However, even though both methadone and heroin have been used for many decades in many countries such as the US, there is still a considerable amount of stigma regarding MOUD which is amplified by tight regulations over its use [30-36]

This is an interesting finding - As a result, a newly developed UHRN SOPs allowed take-home doses of 167 buprenorphine and home induction - you may want to review the old traditional method of diamorphine use in the UK which was self injection at home (see the article as linked earlier). 

We have now added in the discussion that many countries provide less regulated treatment alternatives than in the US. 

It would be helpful if it was clear where quotes start as they are not denoted by italics or quotation marks. I appreciate that this is the journal format and not the authors. 

We have added quotation marks and indenting to make it clearer.

Please ensure that anonymity is preserved in this section and names are changed to ensure participants are not recognised.

We have removed all names from the quotes.   

Your results are quote heavy, with large quotes in the main being the results. It would be better to have a balance of text supported by quotes to amplify a point and bring it to life. Currently it reads as the other way round. You have good information here, so it would be good to discuss your findings and then analyse accordingly. 

We have included a table with the major themes as their representation among participants.  In addition, we now outline the major themes in the results prior to going into more depth with quotes.  We have also edited and shortened many of the quotes.

Each of the CFIR elements has between 2-4 respondents, is this enough to get a flavour of the implementation issues, barriers, influences etc? You had up to 8 respondents to refer to so why is there not greater representation in the results? 

We now include a table which shows the prevalence of each theme among participants.  The quotes selected were emblematic of the themes.

Discussion - I would have hoped to have seen more critical discussion in this section. If the literature was extended in the Introduction this would have been picked up again here to really understand what this study offers by way of intervention for this patient group versus similar initiatives in the world. There is a lot of reported quotes in the results section which can be more targeted and free up words/space for more discussion. 

This is a really interesting study which unfortunately reads as quite descriptive. I do think more can be done to make it a more critical appraisal of what this study shows us about the MAT schemes and its analysis using the CFIR and would urge the authors to reflect on this. 

I would encourage you to introduce new literature in the Introduction section that can be reintroduced to explain your findings. Be more critical in your appraisal of what you did and how it was implemented in the Discussion section and reduce the quote content as it has taken over your results section. 

We now include a paragraph that critically exams pilots of similar programs integrating buprenorphine into community harm reduction services and the importance of CFIR in illustrating this.

Reviewer 2 Report

Dear authors,

I have read your manuscript with great interest, please see below my recommendations. 

1) Introduction: The introduction is complete covering nicely the background around HIV and medication. The aims are clearly presented and prepare the readers for the article content.

2) Methods: Please clarify why there is time variation between the interviews, ranging from 30 minutes to 70 minutes in length.

3) Data analysis: "The first and second 143 authors developed a preliminary coding tree" the author contributions are stated in a different section, usually at the end of manuscript. At this point, its not necessary to know who conducted the coding tree.

4) "People who use drugs in Uganda are considered a social immorality. We haven't yet 191 so much understood drug use as being a disease. We are considering it as people were 192 very spoiled people [who] have chosen their own path in life… In fact we were punishing 193 them for having chosen to use drugs. We consider them black sheep in families and then 194 the other misconception that we have, as Ugandans, is you can wake up one morning and 195 say you're done with using drugs and you actually stop so…they always look at people 196 who use drugs as people who have chosen to live that life… which is not right. …When 197 someone gets addicted… and decided to quit and they quit, they need support. From all 198 Int. J. Environ. Res. Public Health 2021, 18, x FOR PEER REVIEW 5 of 14 entities, they need support from the health workers they need support from the lawyer 199 for us as far as testing them… Most importantly, they need support from their families 200 and their societies so if we can get to a point where you know everyone can accept that 201 fact then I think it will be much easier for us. Truthfully, … before I started working for 202 Harm Reduction Network, I thought it was a matter of choice. (Interview 1) 203 Harm reduction is also very new in Uganda and conflicts with the criminalization of 204 drug use, as the participant below describes."  The particular paragraph is very confusing better to be removed or if its greatly needed rephrase up to the required publication standard.

5) Interview responses not clearly presented, probably include the main points of all responses in one table. I would recommend a specialized software like the NVIVO to show your results.

"One thing you have to appreciate is the fact… [that this] is the first time we have the 217 medical assisted therapy in our country. So that means that it's going to be the first time 218 that we are having buprenorphine being administered from the community level. But also 219 it's going to be the only center of buprenorphine administration from the community. 220 There'll be two centers, that is Butabika which is not so much given out buprenorphine 221 for reasons of the protocol that they have to follow, and then Uganda Harm Reduction 222 Network, so the context is when it's new, it's all new to us, so we shall be doing a little 223 benchmarking, we should be doing a lot of consultation… because we are… are very shy, 224 yes… but the fact that we have people that we can relay… or… consult [if] we are worried 225 about anything, there is no panic so it's very good. (Interview 1)"

Author Response

Thank you for your careful review of our manuscript.  I outline our responses below.

Reviewer 2

1) Introduction: The introduction is complete covering nicely the background around HIV and medication. The aims are clearly presented and prepare the readers for the article content.

Thank you.

2) Methods: Please clarify why there is time variation between the interviews, ranging from 30 minutes to 70 minutes in length.

Interview length varied due to the level of involvement participants had in adapting buprenorphine into the DiC.  We added a statement to clarify this in the methods section.

3) Data analysis: "The first and second 143 authors developed a preliminary coding tree" the author contributions are stated in a different section, usually at the end of manuscript. At this point, its not necessary to know who conducted the coding tree.

We have deleted this statement.

4) "People who use drugs in Uganda are considered a social immorality. We haven't yet 191 so much understood drug use as being a disease. We are considering it as people were 192 very spoiled people [who] have chosen their own path in life… In fact we were punishing 193 them for having chosen to use drugs. We consider them black sheep in families and then 194 the other misconception that we have, as Ugandans, is you can wake up one morning and 195 say you're done with using drugs and you actually stop so…they always look at people 196 who use drugs as people who have chosen to live that life… which is not right. …When 197 someone gets addicted… and decided to quit and they quit, they need support. From all 198 Int. J. Environ. Res. Public Health 2021, 18, x FOR PEER REVIEW 5 of 14 entities, they need support from the health workers they need support from the lawyer 199 for us as far as testing them… Most importantly, they need support from their families 200 and their societies so if we can get to a point where you know everyone can accept that 201 fact then I think it will be much easier for us. Truthfully, … before I started working for 202 Harm Reduction Network, I thought it was a matter of choice. (Interview 1) 203 Harm reduction is also very new in Uganda and conflicts with the criminalization of 204 drug use, as the participant below describes."  The particular paragraph is very confusing better to be removed or if its greatly needed rephrase up to the required publication standard.

We have edited and shortened the quote substantially to make it clearer.

5) Interview responses not clearly presented, probably include the main points of all responses in one table. I would recommend a specialized software like the NVIVO to show your results.

"One thing you have to appreciate is the fact… [that this] is the first time we have the 217 medical assisted therapy in our country. So that means that it's going to be the first time 218 that we are having buprenorphine being administered from the community level. But also 219 it's going to be the only center of buprenorphine administration from the community. 220 There'll be two centers, that is Butabika which is not so much given out buprenorphine 221 for reasons of the protocol that they have to follow, and then Uganda Harm Reduction 222 Network, so the context is when it's new, it's all new to us, so we shall be doing a little 223 benchmarking, we should be doing a lot of consultation… because we are… are very shy, 224 yes… but the fact that we have people that we can relay… or… consult [if] we are worried 225 about anything, there is no panic so it's very good. (Interview 1)"

We now include a table that outlines the major themes and the prevalence of participants who expressed these themes.  We provide a brief description of major themes in the beginning of the results.

Reviewer 3 Report

The manuscript ‘The role of context in integrating buprenorphine into a drop-in center in Kampala Uganda using the Consolidated Framework for Implementation Research’ sheds significant light to the existing research. However, minor improvements are needed.

General comments:

·        Five domains of the CFIR could be could be illustrated by a figure.

·        It is worth distinguishing/ differentiating the text of the quoted statement of the study participants (e.g., in italics)

Detailed comments:

·        I want to note that not all abbreviations have their explanation in the manuscript (e.g., ‘PWUD’ and ‘PWID’), which can be a significant difficulty in understanding the context.

·        Line 227: no title of subsection

·        Punctuation errors, unnecessary spaces or lack of space, etc.

Author Response

Thank you for your careful review of our manuscript.  We outline our responses below.

Reviewer 3

The manuscript ‘The role of context in integrating buprenorphine into a drop-in center in Kampala Uganda using the Consolidated Framework for Implementation Research’ sheds significant light to the existing research. However, minor improvements are needed.

General comments:

  • Five domains of the CFIR could be could be illustrated by a figure.

We now provide a table which includes major themes under each of the CFIR domains and the prevalence of participants who expressed each theme.

  • It is worth distinguishing/ differentiating the text of the quoted statement of the study participants (e.g., in italics)

We have made the quotes clearer by indenting and putting in quotations.

Detailed comments:

  • I want to note that not all abbreviations have their explanation in the manuscript (e.g., ‘PWUD’ and ‘PWID’), which can be a significant difficulty in understanding the context.

We apologize for this oversight.  Acronyms are now spelled out the first time they are used.

  • Line 227: no title of subsection

The subsection was placed their in error and the numbering has been removed.

  • Punctuation errors, unnecessary spaces or lack of space, etc.

We have endeavored to correct all punctuation errors and spacing issues.

Round 2

Reviewer 1 Report

Thank you for your thorough revisions. This version reads much better, well done on your efforts.

Reviewer 2 Report

Dear Authors,

The comments have been addressed and the quality has been improved, therefore I am recommending acceptance of the revised manuscript.

The introduction and presentation of results have been improved, providing a nice flow.